# Hydroxytyrosol–Donepezil Hybrids Play a Protective Role in an In Vitro Induced Alzheimer’s Disease Model and in Neuronal Differentiated Human SH-SY5Y Neuroblastoma Cells

**DOI:** 10.3390/ijms241713461

**Published:** 2023-08-30

**Authors:** Jessica Maiuolo, Paola Costanzo, Mariorosario Masullo, Antonio D’Errico, Rosarita Nasso, Sonia Bonacci, Vincenzo Mollace, Manuela Oliverio, Rosaria Arcone

**Affiliations:** 1Department of Health Science, Institute of Research for Food Safety & Health (IRC-FSH), University Magna Græcia of Catanzaro, Viale Europa, 88100 Catanzaro, Italy; 2Department of Chemistry and Chemical Technologies, University of Calabria, Via P. Bucci, Cubo 12C, 87036 Rende, Italy; 3Department of Movement Science and Well-Being, University “Parthenope” of Naples, Via Medina, 40, 80133 Napoli, Italy; mario.masullo@uniparthenope.it (M.M.); rosaria.arcone@uniparthenope.it (R.A.); 4Department of Health Sciences, University Magna Græcia of Catanzaro, Viale Europa, 88100 Catanzaro, Italyoliverio@unicz.it (M.O.)

**Keywords:** Alzheimer’s disease (AD), donepezil, hybrid compounds, polyphenols, BACE-1, fibrillogenesis, apoptotic death

## Abstract

Alzheimer’s disease (AD) is the most common neurodegenerative pathology among progressive dementias, and it is characterized by the accumulation in the brain of extracellular aggregates of beta-amyloid proteins and neurofibrillary intracellular tangles consisting of τ-hyperphosphorylated proteins. Under normal conditions, beta-amyloid peptides exert important trophic and antioxidant roles, while their massive presence leads to a cascade of events culminating in the onset of AD. The fibrils of beta-amyloid proteins are formed by the process of fibrillogenesis that, starting from individual monomers of beta-amyloid, can generate polymers of this protein, constituting the hypothesis of the “amyloid cascade”. To date, due to the lack of pharmacological treatment for AD without toxic side effects, chemical research is directed towards the realization of hybrid compounds that can act as an adjuvant in the treatment of this neurodegenerative pathology. The hybrid compounds used in this work include moieties of a hydroxytyrosol, a nitrohydroxytyrosol, a tyrosol, and a homovanillyl alcohol bound to the N-benzylpiperidine moiety of donepezil, the main drug used in AD. Previous experiments have shown different properties of these hybrids, including low toxicity and antioxidant and chelating activities. The purpose of this work was to test the effects of hybrid compounds mixed with A*β* _1–40_ to induce fibrillogenesis and mimic AD pathogenesis. This condition has been studied both in test tubes and by an in vitro model of neuronal differentiated human SH-SY5Y neuroblastoma cells. The results obtained from test tube experiments showed that some hybrids inhibit the activity of the enzymes AChE, BuChE, and BACE-1. Cell experiments suggested that hybrids could inhibit fibrillogenesis, negatively modulating caspase-3. They were also shown to exert antioxidant effects, and the acetylated hybrids were found to be more functional and efficient than nonacetylated forms.

## 1. Introduction

Alzheimer’s disease (AD) is the most common neurodegenerative pathology among progressive dementias. To date, there are 47 million people suffering from this disease in the world, and it is estimated that by 2050, the number will be tripled [1,2]. The pathogenesis of AD has been attributed to the accumulation of extracellular aggregates of amyloid protein *β* (A*β*) and neurofibrillary intracellular tangles consisting of τ-hyperphosphorylated proteins in areas of the human brain. Accumulation of A*β* is the consequence of impaired management of *β*-amyloid precursor protein (APP), which is cleaved sequentially by *β*-secretase (BACE-1) and then by the *γ*-secretase complex [3]. APP cleavage generates several beta-amyloid peptides, including A*β* _1–40_ and A*β* _1–42_, which appear to be dominant in the brain. However, A*β* _1–40_ is a more plentiful peptide than A*β* _1–42_ [4]. Under normal conditions, beta-amyloid peptides are synthesized regularly and fall into the cellular physiological metabolism, exerting important trophic and antioxidant properties. On the contrary, a massive presence of these peptides, particularly in unfolded structures, leads to a cascade of events culminating in the onset of AD. A*β*
_1–40_ and A*β* _1–42_ have the propensity to assemble into oligomers and form fibrils that are present in extracellular amyloid plaques in the brain of patients with AD [5]. The fibrillogenic process starts from single monomers of A*β*, which can be associated and form a nucleus (containing six monomers). The set of different nuclei forms a polymer, whose further aggregation leads to the realization of protofibrils, very flexible and stubby, relatively narrow, with a thickness of 5 nm and a length of 150 nm. Further maturation of protofibrils leads to the formation of classical fibrils of A*β*, characterized by a diameter of about 10 nm and an indeterminate length. The presence of fibrils of A*β* has been associated with the pathogenesis of AD, constituting the hypothesis that there is a real “amyloid cascade” [6]. The brain recognizes A*β* plaques as foreign material and triggers an inflammatory and immune response by activating and releasing cytokines. In addition, the presence of high concentrations of A*β* peptides in the central nervous system begins microglial infiltration, worsening the pathological scenario that ultimately leads to cell death and neurodegeneration [7,8,9]. Tau proteins are microneuronal proteins that bind microtubules and are involved in their polymerization and stabilization, maintaining the integrity of the cytoskeleton. This bond is regulated by the phosphorylation of serine and threonine residues operated by a variety of kinases and by cyclin-dependent kinase-5 (CDK5). The A*β* monomers activate CDK5 with subsequent tau phosphorylation. The hyperphosphorylation of these proteins leads to a lower affinity of tau proteins for microtubules. As a result, the structure of the cell can no longer be maintained and, with it, also the synaptic transmission, axonal transport, and signal transduction; the cell undergoes degeneration gradually [10]. These altered proteins, A*β* and τ-hyperphosphorylated, are the precursors to the loss of neuronal connections and eventually the cellular death and loss of brain tissue [11]. However, the risk of occurrence of AD is dependent also on hereditary factors, with more than 40 genetic alterations already identified [12,13]. Typical symptoms of AD include memory loss and difficulties with thinking, language, and problem solving, the progressive decline of activities of daily living, and behavioral and psychological symptoms of dementia [14]. The incidence of dementia increases with age: about 5–8% are affected around the age of 65, while the percentage increases to 70% for patients over the age of 85. In addition, the disease affects more women than men, with a prevalence of 19–29% [15]. Therefore, A*β* accumulation, τ-hyperphosphorylation, and genetic risk factors can be considered responsible for loss of dendritic integrity and progression of disease, although many other factors are involved [16]. To date, there is no specific drug treatment for AD, and only drugs to alleviate the symptoms of the disease are known. The main reason is that the onset of pathological changes related to the disease begins almost a decade before the person shows symptoms. Drugs currently used for AD modulate cholinergic or glutamatergic neurotransmission [17]. The cholinergic system is involved in the cognition process and its dysfunction due to the alteration of the activity of key enzymes such as acetylcholinesterase (AChE) and butyrilcholinesterase (BuChE), responsible for the onset of various dementias including AD [18]. Cholinergic neurons, in which amyloid plaque deposition and neurofibrillary tangles occur, undergo pro-inflammatory events that promote degeneration by altering cognition [19]. In addition, a cholinergic deficiency alters the permeability of the blood–brain barrier, reducing the removal of amyloid plaque and helping to worsen the disease [20]. The dysfunction of the glutamatergic pathway causes memory loss; in fact, glutamate is a key neurotransmitter in maintaining synaptic plasticity. An imbalance in glutamate/glutamine metabolism causes persistent depolarization of neurons, resulting in excitotoxicity and leading to synaptic lesions [21]. Each drug recommended for AD has a different mechanism of action; for example, approved drugs that affect cholinergic transmission are three inhibitors of acetylcholinesterase (AChEI): donepezil, rivastigmine, and galantamine. These drugs are effective in mild to moderate AD and can improve cognition in the patient [22]. For more advanced forms of AD, memantine, an NMDA receptor antagonist that reduces excitotoxicity due to an excess of glutamatergic transmission, was approved in 2003 [23]. To date, due to the lack of pharmacological treatment for AD, chemical research is directed towards the creation of hybrid compounds [24,25] that can achieve three main objectives: (a) modulate cholinergic neurotransmission; (b) inhibit the formation and accumulation of A*β*; and (c) reduce oxidative stress, a common denominator of neurodegenerative diseases. Among the FDA-approved and commonly used drugs against AD, donepezil is the most used for the design of new derivatives, demonstrating a multifunctional activity [26,27,28]. For example, the combination of its pharmacophores group with the natural compounds ferulic acid or curcumin allowed researchers to obtain hybrids of donepezil with robust cholinesterase inhibitory activity [29,30,31]. In this direction, our previous work was based on the design, synthesis, characterization, and biological testing of new donepezil hybrids obtained by association with an olive oil polyphenol such as hydroxytyrosol (HT1, HT1a) and other chemically related compounds HT2, HT3, HT3a, HT4, and HT4a. The hybrid compounds include hydroxytyrosol, nitrohydroxytyrosol, tyrosol, and homovanillyl alcohol moieties bound to the N-benzylpiperidine moiety of donepezil. In particular, several tests were carried out, and the hybrids showed interesting properties and a protective function. In fact, in addition to demonstrating a high structural stability, guaranteed by the effective reproducibility of the results obtained, they excluded a potential cytotoxic effect on SH-SY5Y cells up to the concentration 1–10 μM. To this was added a robust antioxidant effect, exerted by the hybrids in different quantities, on a model of oxidative stress induced in neurons and simulated using H_2_O_2_ as a toxic insult. Antioxidant power has also been demonstrated in vitro using the ORAC and CUPRAC tests. Finally, the catecholic and nitro-group portions were allegedly responsible for the chelating properties [28]. The chemical structure of HT–donepezil hybrids is shown in Figure 1. The purpose of this work is to test the effects of hydroxytyrosol–donepezil hybrids mixed with A*β* _1–40_ to mimic the conditions of AD on a model of neurons SH-SY5Y. In particular, it is essential to understand whether these hybrids can be considered a suitable pharmacological strategy against AD-induced damage.

## 2. Results

### 2.1. Effect of Hydroxytyrosol–Donepezil Hybrids on AChE and BuChE Activity

To determine the effect of hydroxytyrosol–donepezil hybrids on key enzymes involved in AD, such as AChE and BuChE, we performed in vitro enzyme assays in the absence or in the presence of the hybrid. Therefore, to determine the IC_50_ values, the residual enzymatic activity was tested at different concentrations of each molecule (2.5–10.0 µM), and the results are reported in Figure 2. The data, analyzed by a semilogarithmic plot, allowed the determination of the inhibitor concentration required to obtain half-inhibition. The mean values of the IC_50_ calculated based on three different experiments are reported in Appendix A. Regarding cholinesterase activity, the hybrids HT1a, HT3a, and HT4a exhibit a great inhibitory effect on both AChE and BuChE, as indicated by similar and low IC_50_ values. Conversely, the nonacetylated forms, HT1, HT2, HT3, and HT4, showing a less inhibitory activity compared to the respective acetylated ones, appeared to inhibit mostly AChE rather than BuChE.

To assess the type of inhibition, the kinetic parameters of the enzymatic reactions for both AChE and BuChE were determined at different concentrations of the inhibitor. The data, analyzed as reported in the Materials and Methods section, allowed the calculation of the inhibition constant (*K*_i_), reported in Appendix A, and confirmed the behavior obtained using the IC_50_ values. In addition, the data allowed us to derive the mechanisms of inhibition. In particular, on AChE, the inhibitors HT1, HT3, HT3a, HT4, and HT4a acted in an uncompetitive manner (U), whereas the hybrids HT1a and HT2 acted as mixed (M) or noncompetitive (NC) inhibitors, respectively. Regarding the inhibition type determined for BuChE, the data indicated that the majority of the hybrids (HT1, HT1a, HT2, HT3, and HT4a) acted as mixed inhibitors (M), whereas HT3a and HT4 behaved as noncompetitive (NC) inhibitors.

### 2.2. Effect of Hydroxytyrosol–Donepezil Hybrids on BACE-1 Activity

The potential ability of hybrids to act as BACE-1 inhibitors was evaluated by in vitro assays in which the residual enzymatic activity was tested at different concentrations of each inhibitor (2.5–25.0 µM), and the results are reported in Figure 3. The analysis of the data by a semilogarithmic behavior allowed the calculation of the IC_50_ (Appendix A), which indicated that, among the various hybrids, HT1, HT1a, and H3a inhibited BACE-1 activity, showing similar and lower IC_50_ values (about 10–20 µM) compared to those exhibited by the others.

### 2.3. Effect of Hydroxytyrosol–Donepezil Hybrids on Aβ _1–40_ Self-Aggregation

To further characterize the properties of the hybrids on A*β* amyloidogenesis, we investigated whether the molecules could interfere with the process of A*β*
_1–40_ fibrillation. To this aim, the A*β*
_1–40_ peptide was incubated alone or in the presence the HT hybrid, as reported in the Material and Methods section. The results, reported in Figure 4, allowed the calculation of the IC_50_ (Appendix A) and showed that HT1a was the best inhibitor in fibril formation, with an IC_50_ value even lower than that exhibited by donepezil (106.4 ± 11.1 and 183.8 ± 14.7 µM, respectively). HT1 and HT2 showed a similar inhibition activity (375.0 ± 142.8 and 417.6 ± 41.1 µM, respectively) on fibril aggregation, albeit with less efficacy than that of HT1a. The hybrids H3, H3a, H4, and H4a also did not interfere with fibril formation when the molecules were used at even higher concentrations (up to 500 µM).

### 2.4. Effects of Aβ Peptide and Mixtures on Cell Viability

First, we determined the concentration range of A*β* peptide and the time of exposure that did not affect cell viability in our experimental model. The results (Figure 5, panel a) showed that a treatment time of 24 h can be chosen for further investigation, because at the concentrations considered, the higher times (48 and 72 h) determined a reduction in cell vitality of more than 20%. On the contrary, our goal was to improve the early effects generated by treatment with A*β* peptide and not the subsequent ones, when the damage is significant and more difficult to recover from. For these reasons, we chose the concentration of 25 μM of A*β*, which is able to not reduce the cellular viability more than what has been established and, at the same time, not to excessively depart from the concentrations of hybrids chosen in our previous research [28]. In particular, the concentration 0.1 μM of A*β* peptide appears to be ineffective, while we have evaluated a gradual and increasing reduction in cell viability with the concentrations 1, 10, and 100 μM. Figure 5, panel b, shows our evaluation of the cell viability following treatment with the mixtures for 24 h. As can be observed, all mixtures except MIX 1 are able to significantly reduce the damage induced by treatment with A*β* peptide alone.

### 2.5. Antioxidant Property of Mixtures

At an early time, the treatment with MIX 0 induces ROS accumulation; in fact, as shown in Figure 6, panel a, the A*β* peptide causes a significant increase in ROS in the cells after 24 h, but this effect is no longer detectable after 48 and 72 h. Nevertheless, the oxidative damage of A*β* peptide can be detected even after 48 and 72 h; in fact, at longer times, we were able to observe an increasing accumulation of malondialdehyde, a marker of the oxidative stress responsible for lipid peroxidation, which was absent after 24 h.

The treatment with the mixtures was shown to protect, to different degrees, the ROS accumulation induced by the treatment with A*β* peptide. In fact, as can be observed in Figure 7, panel a, the treatment with MIX 0 determined a significant increase in the accumulation of ROS compared to the untreated cells, as demonstrated by the shift on the right of the fluorescent cellular peak. All mixtures were able to reduce this oxidative damage except MIX 1 and MIX 6. However, it is interesting to note that mixtures with their respective acetylated forms (MIX 2 with HT1a; MIX 7 with HT4a) possess protective activities. In general, the mixtures generated by acetylated forms have an antioxidant activity greater than the equivalent not acetylated, demonstrating that acetylation of hybrids improves the effects of the molecule [32]. In these experiments, MIX 4 and MIX 5 worked better than other mixtures, greatly reducing the accumulation of ROS. The respective quantification is shown in panel b.

### 2.6. Aβ Protein Levels in SH-SY5Y Cells

Treatment of SH-SY5Y cells with different mixtures showed a different cytosolic level of A*β* peptide. In particular, Figure 8, panel a, shows the difference in expression of A*β* protein in untreated cells exposed or not to this peptide; in (panel b), the respective quantification is shown. In (panel c), the expression of A*β* in cells treated with several mixtures is shown. The treatment with all the mixtures (except MIX 3) resulted in a significant increase in A*β* peptide level compared to that of untreated cells. However, MIX 5 was more effective than the others, along with MIX 6 and MIX 7 to a lesser extent. The respective quantification is shown in (panel d). In panel e, the amount of A*β* in the absence of cells is shown, and the relative quantification is represented in (f).

### 2.7. Some Hybrid Compounds Prevent Aβ Peptide-Induced Apoptotic Cell Death

In our experimental model, the cells treated with A*β* peptide underwent cell death compared to untreated cells, as shown by the increased expression of caspase 3, usually involved in apoptotic death. The treatment with MIX 1, MIX 2, MIX 3, and MIX 4 reduced the expression of the cleaved form of caspase 3, while MIX 5, MIX 6, and MIX 7 were able to increase it significantly (Figure 9, panels a and b). These results are also confirmed by a cytofluorimetric analysis conducted through the annexin/PI assay, in which it was possible to distinguish the type of death (necrosis or apoptosis). In fact, as shown in Figure 9 (panel c), the treatment with A*β* alone (MIX 0) resulted in reduced viable cells and, at the same time, increased early (Q2) and late (Q3) apoptosis. In addition, a small percentage of necrotic cells (Q4) was also present. On the contrary, the treatment with different mixtures was able to prevent cellular death. In detail, all mixtures except MIX 5 and MIX 7 ensured the reduction in apoptotic death and an increase in viable cells. In (panel d), the quantification is represented.

## 3. Materials and Methods

### 3.1. Materials

Acetylcholinesterase from *Electrophorus electricus* (AChE), butirrylcholinesterase from equine serum (BuChE), acetylthiocholine, butirrylthiocholine, 5′,5′-dithiobis-2-nitrobenzoic acid (DTNB), donepezil, and thioflavine T were purchased from Sigma-Aldrich (Milano, Italy). Human *β*-amyloid peptide (1–40, cat. ab120479) was obtained from Abcam (Cambridge, UK).

Donepezil hybrid derivatives (HT1, HT1a, HT2, HT3, HT3a, HT4, and HT4a) were obtained as previously described [28].

### 3.2. Cholinesterase Assay and Kinetic Analysis

AChE or BuChE activity was assayed by the Ellman method [33] as previously reported [34], using acetylthiocholine or butirrylthiocholine as a substrate, respectively. The enzymatic hydrolysis of thiolated substrates was followed colorimetrically (412 nm) at room temperature (22–27 °C) using a Cary 100 UV–Vis spectrophotometer (Agilent, Santa Clara, CA, USA). The reaction mixture (500 μL) contained 330 μM 5,5′-dithio-bis-2-nitrobenzoic acid (DTNB), 500 μM acetylthiocholine or butirrylthiocholine as substrate, and different amounts of inhibitors in 0.1 M sodium phosphate buffer with pH 7.4. The reaction was started by the addition of 100 mU/mL AChE or BuChE, and the initial rate of the reaction was derived from the linear portion of the kinetics. The concentration of the inhibitor required to reduce the enzymatic activity to 50% (IC_50_) was derived from semilogarithmic plots. Linear curve fits were obtained with the least-squares method, and the significance of the correlation was estimated from the squared correlation coefficient *r*^2^.

The determination of the kinetic parameters *K*_m_ and *V*_max_ for both cholinesterase and monoamine oxidase activities were determined by measuring the initial velocity (*v*_0_) of substrate consumption/product formed at different substrate concentrations, as reported above. In particular, the substrate concentration was in the range of 80–500 µM in the cholinesterase assay and in the 25–150 µM range in the monoamine oxidase assay. Data were nonlinear-fitted to the Michaelis–Menten equation or analyzed by the Lineweaver–Burk equation using Kaleidagraph^TM^ 5.1 software (Synergy, Tokyo, Japan) to derive the kinetic parameters in the absence (*K*_m_ and *V*_max_) or in the presence (*K*’_m_ and *V*’_max_) of the inhibitor. The apparent inhibition constant (*K*’_i_) at each inhibitor concentration was calculated using the following equations:*K*’_i_ = *V*’_max_ · [I]/(*V*_max_ − *V*’_max_) for noncompetitive inhibition (*V*_max_ decrease, *K*_m_ unchanged)(1)
*K*’_i_ = *K*’_m_ · [I]/(*K*’_m_ − *K*_m_) for competitive inhibition (*K*_m_ increase, *V*_max_ unchanged)(2)
*K*’_i_ = *K*’_m_ · [I]/(*K*_m_ − *K*’_m_) for mixed uncompetitive inhibition (*K*_m_ decreased, *V*_max_ decreased)(3)

In the case of mixed inhibition, equations (Equations (1) and (2)) or (Equations (1) and (3)) were used to derive *K*_i_, and the values were mediated. The *K*_i_ values obtained at different inhibitor concentrations were averaged to obtain a unique value of apparent *K*_i_ for each compound. Kinetic parameters and their corresponding standard errors were evaluated using a simple weighting method (Student’s *t* test).

### 3.3. Aβ Self-Aggregation Inhibition Assay

A*β* _1–40_ self-aggregation was obtained by incubating 96 µM peptide in 12 µL of 200 mM sodium phosphate buffer (pH 8.0) containing 0.5% (*v*/*v*) DMSO at 37 °C for 24 h in the absence or in the presence of 24, 120, or 240 µM inhibitor as previously reported [34]. To quantify amyloid fibrils formation, 0.5 mL of 1.6 µM thioflavine T in 50 mM glycine-NaOH buffer (pH 8.5) was added. Therefore, a 300 s time scan of fluorescence intensity was measured using excitation and emission wavelengths of 446 and 490 nm, respectively (slits were set to 10 nm for both the excitation and the emission beams); the fluorescence values at plateau were averaged over a scan of at least 2 min. The percent inhibition due to the presence of the self-aggregation inhibition was calculated from the decrease in the fluorescence signal after the subtraction of the background fluorescence of a thioflavin T solution obtained in the same way. The concentration leading to 50% residual self-aggregation (IC_50_) was derived from a semilogarithmic plot in which the logarithm of the residual self-aggregation was plotted against the inhibitor concentration.

### 3.4. BACE-1 Activity

BACE-1 activity was assayed as previously reported [35,36]. Briefly, the reaction mixtures contained 2.1 ng/µL mouse BACE-1 in 50 mM ammonium acetate buffer, pH 4.5 supplemented with 1 mM triton X-100, and the appropriate amount of the hybrid derivatives. The mixture was incubated for 10 min at room temperature and started by adding 100 nM final concentration of the fluorescent peptide substrate. The increase in fluorescence was followed kinetically using excitation and emission wavelengths of 320 and 420 nm, respectively. The rate was derived from the linear portion of the kinetics, usually in the first 30 min of the reaction. The known BACE-1 inhibitors I and IV were used as positive controls.

### 3.5. Cell Cultures

Human neuroblastoma SH-SY5Y cell line is the most widely used model to reproduce in vitro AD [37]. This cell line was acquired from the American Type Culture Collection (20099 Sesto San Giovanni, Milan, Italy) and maintained in culture in Dulbecco Modified’s Eagle’s Medium (DMEM) (supplemented with 100 U/mL penicillin, 100 µg/mL streptomycin, and 10% thermally inactivated bovine fetal serum). The cells were appropriately differentiated: a treatment with trans retinoid acid 10 µM for 5 days was chosen (Sigma Aldrich, 20151 Milan, Italy). The medium was changed every 2–3 days, and when the cell line reached about 50% confluence, it was supplemented with the mixtures (MIX 0, 1, 2, 3, 4, 5, 6, 7) for a further 24 h. At the end of the treatment, all appropriate tests were performed.

### 3.6. Determination of Cell Viability by MTT Assay

The use of 3-(4,5-dimethyl-2-yl)-2,5-diphenyltetrazole (MTT) bromide allowed the determination of cell viability for the presence of active enzymes within mitochondria in living cells [38]. Mitochondrial esterases can reduce MTT by producing colorimetric modification whose spectrophotometric measurement provides valuable information on cell viability. An amount of 8 × 10^3^ cells/well was plated onto 96-well plates. After 24 h, the culture medium was replaced with fresh phenol-free medium containing a solution of MTT (0.5 mg/mL), and after 4 h of incubation, 100 μL 10% SDS was added to each well to solubilize the formazan crystals. The optical density was measured at wavelengths of 540 and 690 nm using a spectrophotometric reader (X MARK Spectrophotometer Microplate Bio-Rad).

### 3.7. Measurement of Reactive Oxygen Species

The probe used to measure intracellular reactive oxygen species (ROS) is the molecule H_2_DCF-DA [39]. In fact, after its simple diffusion in cells, H_2_DCF-DA is cleaved by cellular esterases, which are responsible for the loss of an acetate group, transforming it into H_2_DCF. This compound can no longer leave the cell and binds the ROS to the highly fluorescent compound DCF. In this way, the quantification of DCF provides the content of cellular ROS. The cells were grown in 96-well microplates with a density of 6 × 10^4^ cells/well and the following day were treated as described. At the end of the treatment, the growth medium was replaced by a fresh, phenol-free medium containing H_2_DCF-DA (25 μM). After 30 min at 37 °C, the cells were washed with PBS, centrifuged, resuspended in PBS, and exposed to H_2_O_2_ (100 μM, 30 min) when required. The fluorescence was evaluated by cytometric analysis (FACS Accury, Becton Dickinson, Milan, Italy).

### 3.8. Measurement of Malondialdehyde (MDA)

Lipid peroxidation was evaluated by measuring the reaction products of MDA with thiobarbituric acid (TBA), which forms a colorimetric product proportional to the MDA present [40]. The cells were placed in 10 cm diameter cell culture and the following day were treated as indicated. At the end of the treatment, the cells were scraped, and then the suspension was subjected to freezing/thawing cycles. Subsequently, 36 mM TBA dissolved in glacial acetic acid was added and the mixture heated for 60 min to 100 °C. The reaction was interrupted by placing the vials on ice for 10 min, and a spectrophotometric reading was carried out at 532 nm.

### 3.9. Preparation of Whole-Cell Protein Lysates and Immunoblot Analysis

The total lysates of SH-SY5Y cells were obtained from 6-multiwells. Lysates were taken after the cells were exposed to a preheated lysis buffer (80 °C) containing 50 mM of Tris-HCl (pH 6.8), 2% of SDS, and a protease inhibitor mixture and immediately boiled for 2 min. The protein concentration was determined using a DCA protein assay. After the addition of 0.05% bromophenol blue, 10% glycerol, and 2% *β*-mercaptoethanol, the samples were boiled again and loaded onto 12% SDS-polyacrylamide gels. Following electrophoresis, the polypeptides were transferred to nitrocellulose filters and blocked with TTBS/milk (TBS 1%, Tween 20, and nonfat dry milk 5%), and then the antibodies were used to reveal the respective antigens. Primary antibodies were incubated overnight at 4 °C, and, after washing, the filters were exposed to horseradish peroxidase-conjugated secondary antibody for 1 h at room temperature. The blots were developed using the chemiluminescence procedure. Experiments of immunoblot analysis were carried out in two ways: (a) electrophoresis of proteins obtained from cell extracts, as described; and (b) electrophoresis of proteins obtained by the formation of mixtures in the test tubes (in the absence of cells). The following primary antibodies were used: a rabbit monoclonal antibody for cleaved caspase 3 (Ab 9H19L2, Invitrogen, at dilution 1:1000), a mouse monoclonal antibody for *β*-amyloid, (Ab11132, Abcam, at dilution 1:1000), and a mouse monoclonal antibody for actin (Sigma Aldrich, at 1:5000 dilution). Horseradish peroxidase-conjugated goat anti-mouse and anti-rabbit were used as the secondary antibodies at 1:10,000 dilution.

### 3.10. Annexin V Staining

Cells at the concentration of 1 × 10^6^ cells/ ml were appropriately treated, trypsinized, washed with cold PBS, and resuspended in a specific buffer (Annexinv/ Kit for apoptotic death). Subsequently, 5 mL of FITC Annexin V (BD Biosciences, San Jose, CA, USA) was added, and the samples were gently vortexed and incubated for 15 min at 25 °C in the dark. The cytofluorimetric analysis (emission filter 515–545 nm for FITC; 600 nm for PI) was performed on 30,000 cells per sample using a FACS Accuri laser flow cytometer (Becton Dickinson, Milan, Italy).

### 3.11. Preparation of Aβ–Hybrid Mixtures

A*β* 25 μM peptide was incubated alone or in the presence the specific amount of each hybrid compound [28] for 24 h at room temperature prior to the cell treatment. The cells were then exposed to these mixtures for 24 h for the analysis of the specific effect. A summary diagram of the mixtures’ preparation is shown in Figure 10.

## 4. Discussion

As already described, one of the pathological characteristics of AD is generated by the accumulation of the A*β* protein that determines the aggregation and formation of fibrils, which are responsible for the alteration of the functioning of synapses. This phenomenon leads to cognitive decline and degeneration of large areas of the cerebral cortex [41]. Under physiological conditions, A*β* possesses a specific physiological function; however, factors such as gene mutations, aging, or oxidative stress are responsible for the alteration of A*β* homeostasis and its accumulation in the form of oligomers and fibers deposited in the brain, causing neurotoxic damage, the functional death of the neuron, cognitive impairment, and dementia [42]. It is important to note that A*β* in fibrils is a consequence of protein misfolding and is notoriously linked to a neurodegenerative disorder or to folding diseases; its conformation is in contrast with the native, well-folded conformation [43]. To establish in our experimental model the aggregation of A*β* and the formation of fibrils, characterizing AD, we set up experimental conditions in vitro using a method already reported [44]. The first important factor is the choice of the concentration of A*β* to be used in the test tube; actually, higher concentrations are needed than those present in physiological settings. This condition is known as “supersaturation” and, since the physiological concentration of A*β* is in the order of sub-nM, we chose to use a higher concentration in the μM range [45]. To this aim, we chose the not-cytotoxic concentration of 25 µM A*β* (Figure 6). In vitro aggregation kinetics of A*β* is a process typically divided into three phases: an initial lag phase, a subsequent growth phase, and a final stationary phase, where mature fibrils can be obtained from A*β* monomers [46]. In the search for multitarget inhibitors for the treatment of AD, we checked the effect of some hydroxytyrosol derivatives of donepezil on the activity of enzymes targets of pharmacological action, such as AChE, BuChE, and BACE-1 as well as on the fibril formation of the A*β* peptide. The hybrid compounds tested exhibited a very similar inhibition power of AChE, which was lower than that exerted by donepezil (Figure 3 and Appendix A); conversely, a different behavior was observed regarding the inhibition power towards BuChE. In fact, among the HT hybrids, the higher inhibition power was observed for the acetylated form, and this finding led to a reduced selectivity between AChE and BuChE inhibition. Concerning BACE-1 inhibition, a similar behavior to BuChE was found, pointing to a higher inhibition power exerted by the acetylated hybrids forms, with two exceptions concerning an acetylated form (HT4a) and the not-acetylated one (HT1) showing lower and higher inhibition power, respectively (Figure 4 and Appendix A). Regarding the effect of the hybrid compounds on the A*β* fibril formation process, only HT1, HT1a, and HT2 exerted an inhibitory activity, with HT1a being the most powerful inhibitor to a level higher than that displayed by donepezil (Figure 5 and Appendix A). Subsequently, we tested the effects exerted by the hybrid compounds on the A*β* behavior using the neuronal SH-SY5Y cells mimicking an AD model. For this purpose, we preincubated the HT–donepezil hybrids with A*β* 25 μM for 24 h to obtain mixtures that were used for in vitro assays. Treatment with MIX 0 induced net oxidative damage, as already demonstrated in several districts [47]. Oxidative damage was evident after 24 h of treatment, when accumulation of ROS occurred; in prolonged times, no ROS were detected, but there was the appearance of malondialdehyde, responsible for lipid peroxidation (Figure 7). Our results indicated that some HT–donepezil hybrids are able to protect against the A*β*-induced ROS accumulation. Polyphenols, compounds naturally present in different foods of plant origin, possess numerous beneficial properties for human health including antioxidant, anti-inflammatory, and anticancer properties, among others, slowing the development of several diseases such as cardiovascular, neurodegenerative, and uncontrolled proliferation diseases [48,49]. The biological activity of polyphenols is closely related to their antioxidant properties, as they were able to reduce reactive oxygen species. To date, the polyphenols of olive oil have been found to be particularly protective against various pathologies [50]. Among all these compounds, we chose to build hybrids from hydroxytyrosol, which has been shown to have excellent antioxidant activity [51,52]. The treatment with all mixtures except MIX 1 and MIX 6 showed an antioxidant effect compared to treatment with MIX 0. In addition, the mixtures containing hybrids in acetylated forms could reduce the accumulation of ROS more than mixtures with nonacetylated hybrids. In fact, MIX 2 and MIX 7 had a much greater and more significant protective effect than MIX 1 and MIX 6, respectively. MIX 5 was also more effective than its counterpart MIX 4, although the values corresponding to the accumulation of ROS were much more similar (Figure 8). Evaluation of the effect of mixtures on the cell line gives different results, presumably due to the metabolic action conducted by the cells [53]. An example is shown in Figure 9, where we can compare the expression of A*β* when mixtures were administered to cells. In the absence of cells and after the mixing time of A*β* peptide with HT–donepezil hybrids (24 h), which ensured the formation of fibrils, the expression of A*β* appeared mostly identical in all samples examined. Conversely, the presence of cells showed a modulation in the expression of this protein. In particular, A*β* cytosolic increased following exposure to all the mixtures compared to that obtained with A*β* alone. In this case, the acetylated forms of the HT–donepezil hybrids resulted in a further increase in the cytosolic level of A*β* peptide compared to their nonacetylated counterparts. If this interpretation is true, we suggest that the A*β* cellular level increased after exposure to all the mixtures. The variability observed could be due to the different efficacies of the molecular interaction between the hybrids and A*β*. Therefore, we can suggest that HT–donepezil hybrids exert a protective role due to their ability to decrease the damage induced by amyloid fibrils in SH-SY5Y cells [54]. Moreover, the acetylated forms of hybrids (MIX 2, MIX 5, MIX 7) showed a better result than nonacetylated ones. A similar interpretation could also be used to explain the results of possible involvement of apoptotic death. Once again, the expression of caspase-3 is reduced with MIX 1, MIX 2, and MIX 3, just as it was for the expression of A*β*. Furthermore, the hybrids containing an acetylated moiety (MIX 5 and MIX 7), except for MIX 2, seemed to trigger a greater involvement of caspase-3 than the mixtures containing nonacetylated forms. The development of caspase-3 expression reflects also the results obtained from the annexin-PI assay. In fact, MIX 1, MIX 2, and MIX 3 reduced late apoptosis (Q3) and increased early apoptosis (Q2), an effect that could result from the reduction in caspase-3 expression and from a lower ability of these mixtures to reduce fibrillogenesis. In addition, mixtures containing HT–donepezil acetylated hybrids (MIX 5 and MIX 7) substantially increased late apoptotic death, almost equaling treatment with A*β* alone. If HT–donepezil hybrids were able to reduce fibrillogenesis, it would increase the harmful effect of A*β* monomers while reducing the severity generated by fibrils. It should be noted that A*β* is also neurotoxic; in general, it is an amphipathic peptide with 16 polar and 12 nonpolar amino acids (12 are charged at neutral pH). This feature promotes the insertion of the protein into the membranes. The insertion of this peptide could disrupt and destabilize the structure and function, altering the curvature of the membrane surface and interrupting the symmetry of the fatty acyl chains [55].

All together, these data suggest the following:
(a)HT–donepezil hybrids are able to provide protection against oxidative effects;(b)Acetylation has made HT–donepezil hybrids more functional and efficient;(c)Some mixtures seem to be able to inhibit fibrillogenesis, negatively modulating caspase-3 and apoptotic death.


## 5. Conclusions

Our knowledge and theoretical interpretations of HT–donepezil hybrids are still at an early stage and require numerous clarifications and confirmations. Therefore, further future studies will be directed towards understanding the mechanisms involved. In our system model, the hybrid compounds used showed protective action against A*β*-induced cell toxicity. Therefore, they can be considered lead compounds as potential drugs to be used against AD pathology.

## Figures and Tables

**Figure 1 ijms-24-13461-f001:**
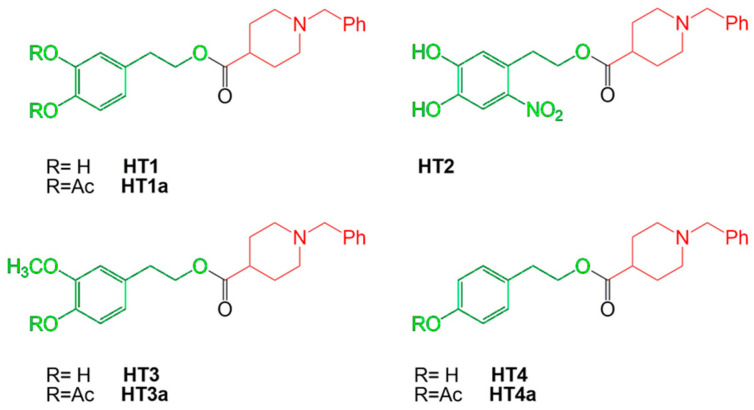
Chemical structures and nomenclature of HT–donepezil hybrids. The N-benzylpiperidine moiety of donepezil and the hydroxytyrosol derivative moieties are indicated in red and green, respectively.

**Figure 2 ijms-24-13461-f002:**
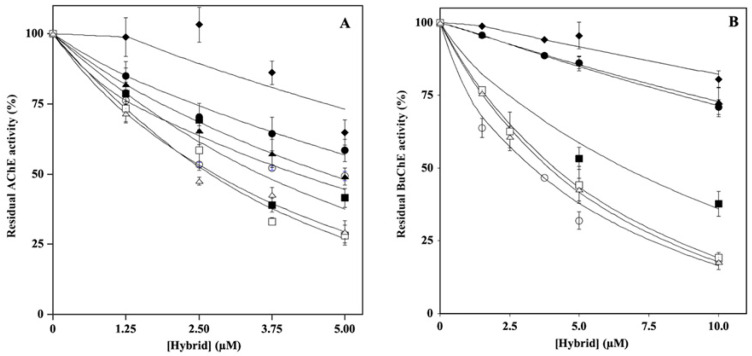
Effect of HT hybrids on cholinesterase activity. The residual AChE (**A**) or BuChE (**B**) activity was determined in the absence or in the presence of the indicated concentration of HT1 (filled circles), HT1a (empty circles), HT2 (filled lozenges), HT3 (filled squares), HT3a (empty squares), HT4 (filled triangles), and HT4a (empty triangles), as reported in the Material and Methods section. Data points are shown as mean percentage values ± sd calculated with respect to the activity measured in the absence of the hybrid on at least 6 different determinations.

**Figure 3 ijms-24-13461-f003:**
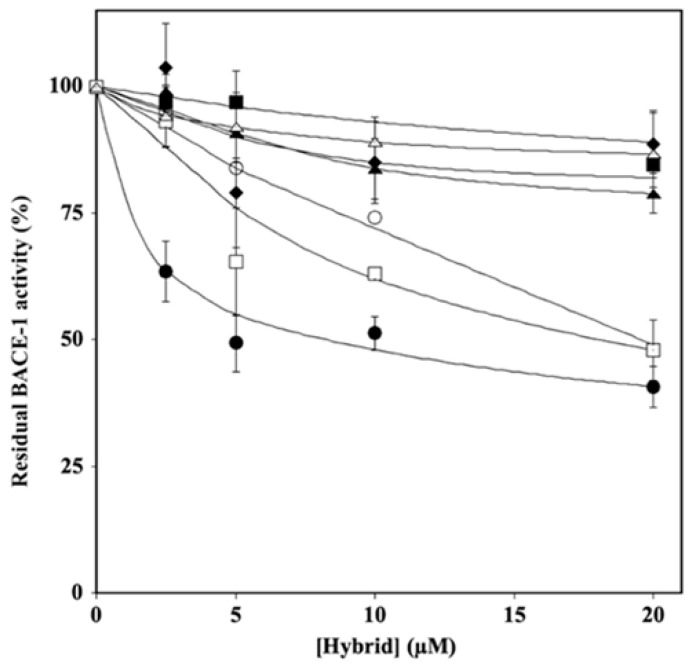
Effect of HT hybrids on BACE-1 activity. The residual BACE-1 activity was determined in the absence or in the presence of the indicated concentration of HT1 (filled circles), HT1a (empty circles), HT2 (filled lozenges), HT3 (filled squares), HT3a (empty squares), HT4 (filled triangles), and HT4a (empty triangles), as reported in the Material and Methods section. Data points are shown as mean percentage values ± sd calculated with respect to the activity measured in the absence of the hybrid on at least 3 different determinations.

**Figure 4 ijms-24-13461-f004:**
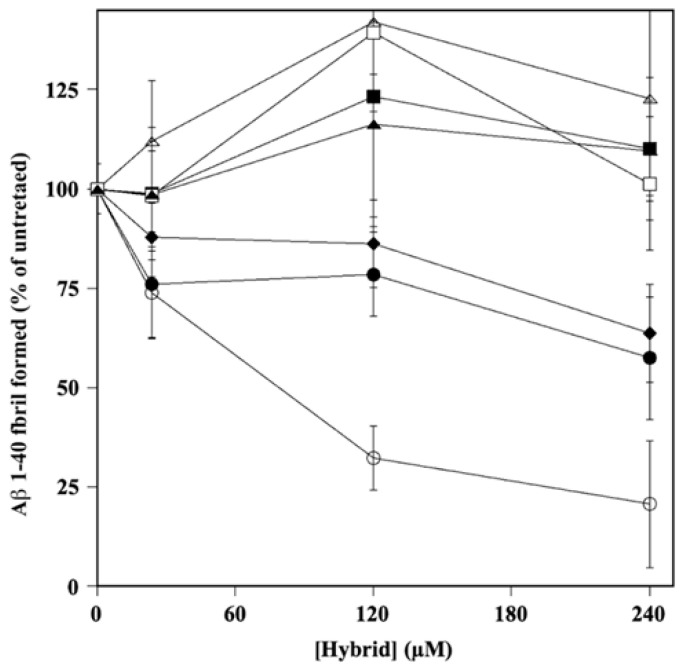
Effect of HT hybrids on A fibril formation. The amount of A 1–40 fibril formation was determined in the absence or in the presence of the indicated concentration of HT1 (filled circles), HT1a (empty circles), HT2 (filled lozenges), HT3 (filled squares), HT3a (empty squares), HT4 (filled triangles), and HT4a (empty triangles), as reported in the Material and Methods section. Data points are shown as mean percentage values ± sd calculated with respect to the activity measured in the absence of the hybrid on at least 3 different determinations.

**Figure 5 ijms-24-13461-f005:**
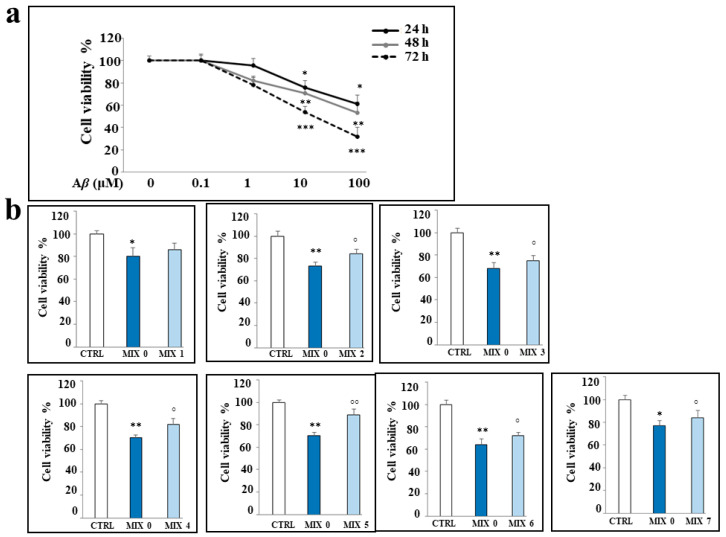
Effects of the treatments with A*β* peptide and/or different mixtures on viability of SH-SY5Y cells. (**Panel a**): cell viability following exposure to A*β* peptide (0.1–100 μM) for 24, 48, and 72 h. (**Panel b**): cell viability following the treatment with A*β* peptide 25 μM alone (MIX 0) or in combination with the different mixtures for 24 h. Three independent experiments were carried out, and the values are reported as mean ± sd. * denotes *p* < 0.05 vs. the control; ** denotes *p* < 0.01 vs. the control; *** denotes *p* < 0.001 vs. the control. ° denotes *p* < 0.05 vs. A*β*; °° denotes *p* < 0.01 vs. A*β*. Variance analysis (ANOVA) was followed by a Tukey–Kramer comparison test.

**Figure 6 ijms-24-13461-f006:**
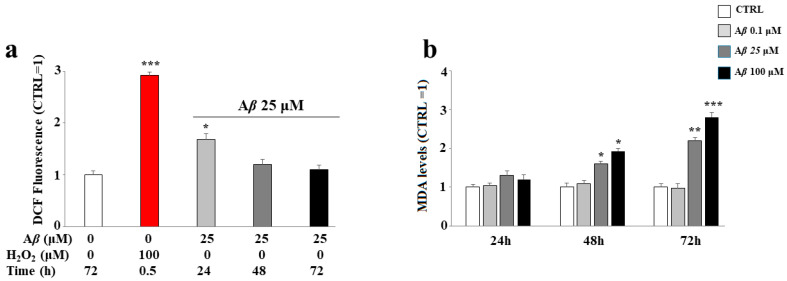
Effect of the treatment with A*β* peptide on oxidative damage in SH-SY5Y cells. Cells were treated with the A*β* peptide for the times and the conditions indicated. ROS (**panel a**) and malondialdehyde (**panel b**) levels were determined as reported in the Material and Methods section. In panel a, hydrogen peroxide was used as positive control. Three independent experiments were carried out, and the values were expressed as mean ± sd. * denotes *p* < 0.05 vs. the control; ** denotes *p* < 0.01 vs. the control; *** denotes *p* < 0.001 vs. the control. Student’s two-tailed *t* test was performed.

**Figure 7 ijms-24-13461-f007:**
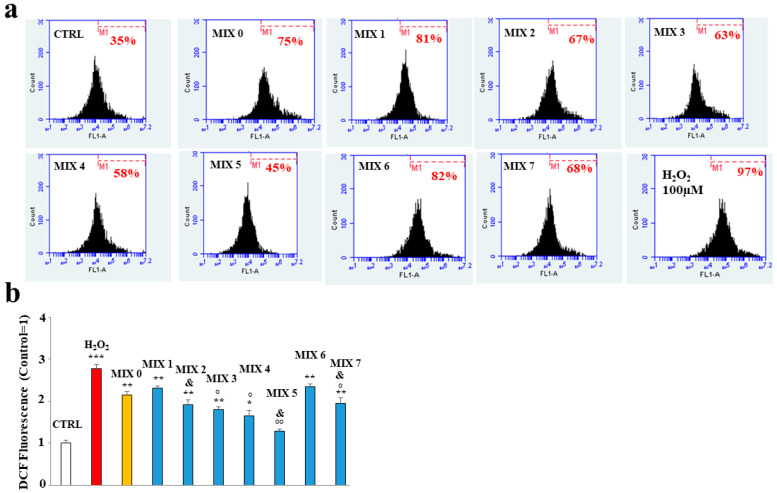
Effect of the hybrid compounds on oxidative damage in SH-SY5Y cells determined by cytofluorimetric analysis. In (**Panel a**), the boxes represent the different treatments. In each box is shown a marker (M1), which has been arbitrarily chosen for untreated cells and has been kept identical for all treatments. The number of cells included in M1 is expressed as a percentage in each box. Three independent experiments were carried out with the same trend, and an indicative experiment is shown. (**Panel b**) reports the quantification calculated from the comparison of percentages and reported as fold-change vs. CTRL cells. Above each histogram, the relative treatment is indicated, and different colors have been used to represent several treatments. In particular, untreated cells (CTRL) are represented in white, the positive control (H_2_O_2_) in red, the mixture containing only A*β* peptide (MIX 0) in orange, and, finally, all other mixtures in blue. The values are expressed as mean ± sd. * denotes *p* < 0.05 vs. the control; ** denotes *p* < 0.01 vs. the control; *** denotes *p* < 0.001 vs. the control. ° denotes *p* < 0.05 vs. MIX 0; °° denotes *p* < 0.01 vs. MIX 0. & denotes *p* < 0.05 vs. the respective MIX with a nonacetylated hybrid. Variance analysis (ANOVA) was followed by a Tukey–Kramer comparison test.

**Figure 8 ijms-24-13461-f008:**
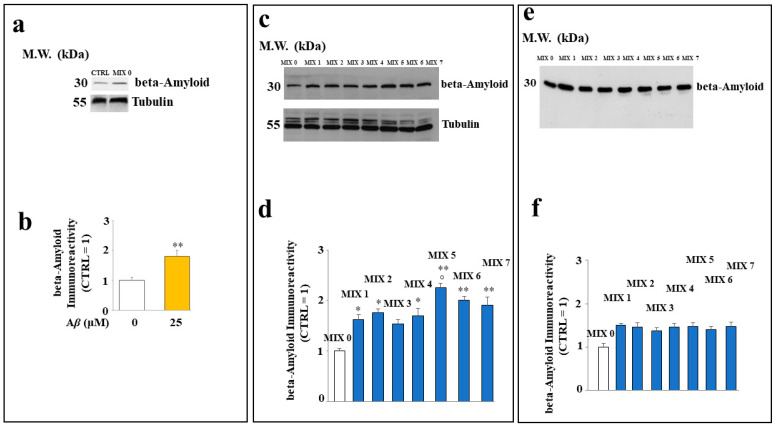
Effect of the hybrid compounds on the cytosolic A*β* protein level in SH-SY5Y cells. Western blotting analysis of total protein extracts of cells untreated or exposed to A*β* (**Panel a**) or to the different mixtures (**Panel c**). (**Panels b**,**d**) report the respective quantification. To compare the expression of A*β* in the presence or in the absence of cells, Western blotting experiments were also carried out on samples incubated in test tubes, as shown in (**Panels e**,**f**). Three independent experiments were carried out, and the values are expressed as mean ± sd. * denotes *p* < 0.05 vs. the control; ** denotes *p* < 0.01 vs. the control. ° denotes *p* < 0.05 vs. the respective MIX with a nonacetylated hybrid. Variance analysis (ANOVA) was followed by a Tukey–Kramer comparison test.

**Figure 9 ijms-24-13461-f009:**
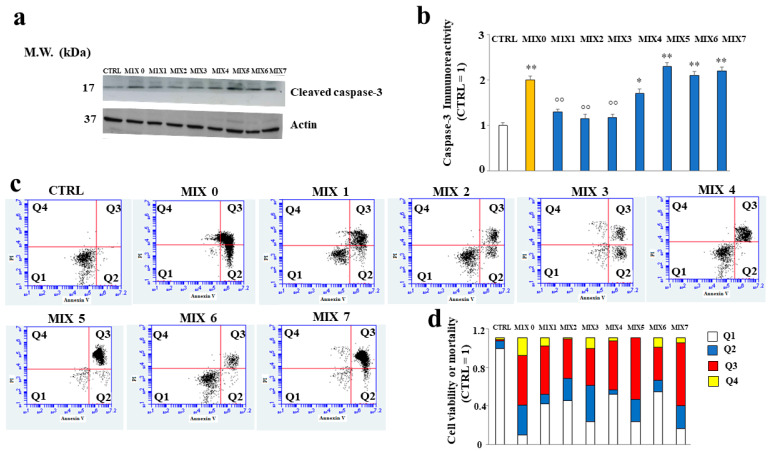
Involvement of cleaved caspase-3 and mortality assessed by Annexin V/PI staining assay. In (**Panel a**), the expression of cleaved caspase-3 is represented, and in (**Panel b**), the respective quantification is shown. Three independent experiments were carried out, and the values are expressed as mean ± sd. * denotes *p* < 0.05 vs. the control; ** denotes *p* < 0.01 vs. the control. °° denotes *p* < 0.01 vs MIX 0. Variance analysis (ANOVA) was followed by a Tukey–Kramer comparison test. (**Panel c**) shows cytometric analysis conducted on 30,000 cells; each treatment is represented through a dot plot divided into 4 quadrants (Q1, Q2, Q3, and Q4). Q1 refers to Annexin V-negative/PI-negative cells (viable cells). Q2 refers to Annexin V-positive/PI-negative cells (early apoptotic cells). Q3 refers to Annexin V-positive/PI-positive cells (late apoptotic cells). Q4 refers to Annexin V-negative/PI-positive cells (advanced necrosis). A representative experiment of three independent experiments is shown. In (**Panel d**), the expression of results obtained by Annexin V/PI staining assay is shown.

**Figure 10 ijms-24-13461-f010:**
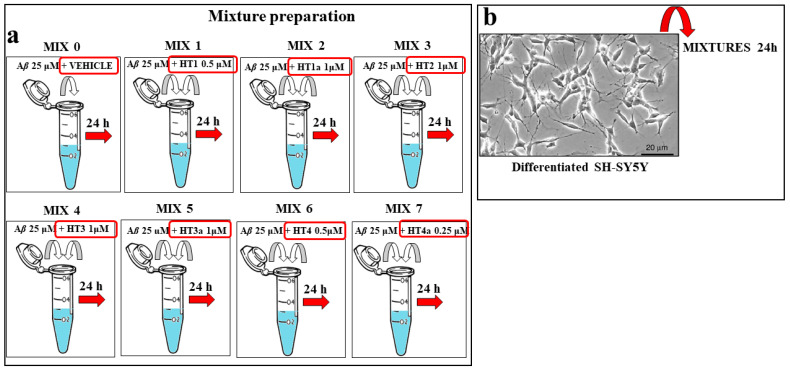
Experimental protocol for the mixture preparation of hybrid compounds and A*β*. (**Panel a**) Scheme of the preparation of mixtures. Each hydroxytyrosol–donepezil hybrid (HT1, HT1a, HT2, HT3, HT3a, HT4, and HT4a) has been mixed with the A*β* 25 μM (MIX 1, MIX 2, MIX 3, MIX 4, MIX 5, MIX 6, and MIX 7) and kept them together for 24 h in order to form the amyloid fibrils. The MIX 0 consisted of the A*β* 25 μM peptide in the presence of the respective vehicle. (**Panel b**) Differentiated SH-SY5Y cells were treated with MIX 0 or with the prepared mixtures for 24 h. Untreated cells were used as internal control.

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
