# Peer review of "Hydroxytyrosol–Donepezil Hybrids Play a Protective Role in an In Vitro Induced Alzheimer’s Disease Model and in Neuronal Differentiated Human SH-SY5Y Neuroblastoma Cells"

_ijms, 2023, doi:10.3390/ijms241713461_

Round 1
Reviewer 1 Report
This study has used HT-donepezil hybrid compounds to mix with Aβ 1-40 to induce fibrillogenesis and mimic AD pathogenesis. There are many results from multiple tests and authors found that those hybrids could inhibit fibrillogenesis, negatively modulating caspase-3 and able to provide protection against oxidative effects. That’s interesting finding. But just as authors mentioned in the conclusion by themselves, this work is still at an early stage and require numerous clarifications and confirmations.
Major concerns:
1. The font of the text is not uniformed, and abstract words are different from others. It seems that the manuscript was not carefully checked and corrected before the submission. And there is even a comment included in the doc. File. Please make careful correction and revision before submitting any manuscript. Do not make it looks like an incomplete manuscript.
2. The table 1 and table 2 in the results part are very difficult for understanding. Why does the work use the same concentration of Ab in each mix but with different concentrations of different HTs? It is very confusing if the author would like to claim that they have investigated the concentration effects by showing either of the tables. It would be much easier for readers to know what you are addressing here by showing graph with curves or bars.
3. In Fig 3, a showed cell viability at 24, 48 and 72h. But which time point results have been shown in panel b? And the significance and mean values have been described without including the n. For comparing to control only, why choose ANOVA but not t-test? Please explain or re-do it. The same as Fig 4. Please also explain there.
4. Fig 5, how many repeat have you done? You have use different color bars, please also explain the meaning of color in figure legends.
5. In 3.7, the author claims “Hybrid compounds prevent Aβ peptide induced apoptotic cell death.” But the results clearly showed not all HTs prevent Aβ peptide induced apoptotic cell death even Mix5 and Mix7 had facilitated apoptosis. How to explain the conflicts here?
6. In Fig 8, the author was trying to show the effect of hybrid compounds on the penetration of Aβ into SH-SY5Y cells by showing the immunofluorescent images. But how to define Aβ peptide cell localization without any nuclear or membrane marker? And almost all different HTs treated showed different Aβ staining effect. Is it due to different brightness/contrast of the image, or different scanning laser power? Anyone wants to compare signal brightness of immunofluorescence among different sample, please show average intensity or total intensity after normalization for the comparison.
Minors,
Please remove the part of classification of olive oil polyphenols in the discussion. It is non-relevant to discussion. Cite references would be already too much.
Author Response
Dear reviewer, thank you for your valuable suggestions. Attached you will find our answers to your suggestions. In the file, your comments are highlighted in yellow, while our answers are highlighted in green.

Reviewer 2 Report
The authors investigated the hydroxytyrosol-donepezil hybrids' effect on Aβ 1-40 aggregation in the neuronal cell line SH-SY5Y. Initially, the hydroxytyrosol-donepezil hybrids were evaluated for their AchE and dBuChE activity, and it found that most of the hybrids exhibited good inhibitor activity towards AchE and BuChE and also showed some effect on BACE-1 activity. Only HT1a, HT1, and HT2 exhibited significant inhibition of fibril formation—the authors next tested Aβ fibrilization with no cell viability and ROS activity in the SH-SY5Y cell line. Finally, the authors did various tests to evaluate Aβ fibrilization and its effects on cells with the hybrid compounds mixed with Aβ in SH-SY5Y. They found some significance in the change of Aβ accumulation and aggregation within the cells.
One strength of the paper is that the authors investigated various parameters comprehensively with well-established negative and positive control small molecules. Furthermore, the authors selected various uncompetitive, mixed, or non-competitive inhibitors of AchE. Furthermore, their hybrid derivatives did not have as strong an effect on BACE-1 activity. While not exhibiting activity, the authors further tested their activity and saw that BACE-1 inhibition is a requirement of their hybrid for activity. One of the indicators was ROS damage in proteins and lipids. The values obtained by the authors represent significance.
One weakness of the paper is that compounds that exhibit selectivity towards AchE over BASE-1 were not further evaluated. Another weakness is that the authors require a very high inhibitor concentration to prevent Aβ aggregation. The requirement for higher concentration seems to affect other experiments downstream where the effect is not as pronounced. In many cases, MIX0 has very similar activity to the previous mixes, and a deeper investigation of cooperative scoring would be required. Finally, one major issue is SH-SY5Y. While it is a typical enough cell line for evaluation, it still is an immortalized cell line derived from glioma that could have significantly different activity than primary neuronal cells.
Overall, the authors did an excellent job of investigating these small molecule hybrids and characterizing their effects within cells. While the significance could be improved, it sets the groundwork for further investigation into cellular trafficking, specifically the role of AchE and BuChe in Aβ aggregation. Furthermore, presenting a small molecule able to induce vocalization and excretion of Aβ is an important discovery that could be further improved upon by future researchers. I suggest that the paper be accepted into your journal following minor revisions.
Author Response
Dear reviewer, thank you for your valuable suggestions. Attached you will find our answers to your suggestions. In the file, your comments are highlighted in yellow, while our answers are highlighted in green

Round 2
Reviewer 1 Report
This work has done a series of experiments with abundant results of hybrid compounds effects with Aβ on AD pathogenesis. It has offered more insight on the pharmacological strategy against AD induced cell death.
The authors have well addressed all my comments and done the revision correspondently in the resubmitted manuscript. I suggest to accept it in present form.